# Modulation of the Cellular microRNA Landscape: Contribution to the Protective Effects of High-Density Lipoproteins (HDL)

**DOI:** 10.3390/biology12091232

**Published:** 2023-09-13

**Authors:** Annette Graham

**Affiliations:** Department of Biological and Biomedical Sciences, School of Health and Life Sciences, Glasgow Caledonian University, Cowcaddens Road, Glasgow G4 0BA, UK; ann.graham@gcu.ac.uk; Tel.: +44-(0)141-331-3722

**Keywords:** high-density lipoproteins, gene regulation, signal transduction, microRNA biogenesis, microRNA secretion

## Abstract

**Simple Summary:**

High-density lipoproteins are often described as the ‘good’ cholesterol in the bloodstream, as this complex of lipids (fats) and proteins is known to protect many cells and tissues against damage. One aspect of this protective function may be the ability of this lipoprotein class to modify the expression of small pieces of RNA (microRNA) which can regulate the expression of networks of genes which encode proteins that mediate cellular functions. Understanding the sequences involved in sustaining cellular function may provide a novel route to new therapeutics.

**Abstract:**

High-density lipoproteins (HDL) play an established role in protecting against cellular dysfunction in a variety of different disease contexts; however, harnessing this therapeutic potential has proved challenging due to the heterogeneous and relative instability of this lipoprotein and its variable cargo molecules. The purpose of this study is to examine the contribution of microRNA (miRNA; miR) sequences, either delivered directly or modulated endogenously, to these protective functions. This narrative review introduces the complex cargo carried by HDL, the protective functions associated with this lipoprotein, and the factors governing biogenesis, export and the uptake of microRNA. The possible mechanisms by which HDL can modulate the cellular miRNA landscape are considered, and the impact of key sequences modified by HDL is explored in diseases such as inflammation and immunity, wound healing, angiogenesis, dyslipidaemia, atherosclerosis and coronary heart disease, potentially offering new routes for therapeutic intervention.

## 1. Introduction

This narrative review article sets out to explore the impact of high-density lipoproteins (HDL) on the cellular microRNA landscape, and to assess the attribution of these changes to the protective effects of HDL reported to sustain cellular function. Historically, epidemiological studies have posited that HDL-cholesterol (C) levels are inverse to the risk of cardiovascular disease (CVD), but some recent genetic and pharmacological studies have questioned the linear nature of this relationship [1,2,3]. While low levels of HDL-C are clearly related to raised risk of CVD, the inverse cannot be established for elevations in HDL-C; indeed, the latter have been variously linked with non-vascular diseases, including cancer [4,5]. A U-shaped association between HDL-C and all-cause mortality has been suggested, with increased risk noted at both extremely high and low HDL concentrations [5,6]. A Mendelian randomisation (MR) study, which examined the single nucleotide polymorphisms (SNPs) of endothelial lipase, questioned the vascular benefits ascribed to HDL [7], although assessment of the genetic and secondary causes of chronic HDL deficiency indicated a higher prevalence of CVD [8]. More recently, the application of multi-variant MR and MR-Egger methodologies, combined with analysis of datasets derived from large genome-wide association studies (GWAS), have indicated the protective effects of HDL-C in reducing the risk of CVD and diabetes [9,10]. The benefits of HDL-C-raising therapeutic strategies have also proved hard to establish conclusively, with the use of nicotinic acid, cholesteryl ester transfer protein (CETP) antagonists, fibrates and dietary interventions providing variable outcomes in clinical studies (reviewed in [1]).

These findings led to the development of a distinct focus on the complex concept of HDL quality (rather than quantity) and functionality [11,12]. Thus, while the quantity of HDL is routinely defined as serum HDL-(cholesterol) concentration (mg/dL), HDL quality is thought to be determined by the content of protein, lipid and other cargo molecules, and its function indicated by the impact of this lipoprotein on cholesterol efflux, antioxidant, antithrombotic and anti-inflammatory activities [12].

### 1.1. HDL: Composition and Cargo

High-density lipoproteins are highly heterogeneous in terms of their physical, chemical and biological properties, and divided into several subclasses which vary in density, size, shape and lipid and protein composition (reviewed in [13]). The proteome of HDL is complex (>200 proteins), and evolving (https://homepages.uc.edu/~davidswm/HDLproteome.html, accessed on 5 July 2023), with differences reported that depend on the methodologies utilised to isolate the HDL fraction(s) [13,14]. In brief, the protein components variously include apolipoproteins, such as apoA-I, apoA-II and apoE, enzymes (e.g., lecithin cholesterol acyl transferase [LCAT], and paraoxonase [PON]), and lipid transfer proteins (CETP and phospholipid transfer protein [PLTP]) as well as complement components, proteinase inhibitors, acute phase response proteins and glycoproteins [13]. Importantly, these proteins are not uniformly distributed among the various subclasses of HDL, with key proteins being enriched in denser (HDL3a, HDL3b, HDL3c) and lighter (HDL2a, HDL2b) fractions isolated by isopycnic density gradient ultracentrifugation; the proteome within HDL also accommodates differing apolipoprotein isoforms, and undergoes translational and post-translational modifications. Intriguingly, epidemiological studies suggest that perturbation of the HDL proteome—characterised by loss of lipoprotein phospholipase A_2_ (Lp-PLA2), glutathione peroxidase (Gpx-3), PON1 and PON3, and gain of prooxidant and proinflammatory mediators (myeloperoxidase and serum amyloid A)—is linked with diabetes [15,16], CVD, SARS-CoV-2 (severe acute respiratory syndrome coronavirus 2) infection, renal disease [17] and Alzheimer’s disease [18].

The lipidome (>200 species) of HDL has been carefully characterised using FPLC and mass spectrometric analyses [19], with phosphatidylcholine (PC) proving the major phospholipid present, which is distributed evenly among HDL subclasses. The degradation product of PC, lysophosphatidylcholine (LPC) is also a major phospholipid found in HDL, together with phosphatidylethanolamine and other glycerophospholipids, plasmalogens, cardiolipin, sphingolipids and neutral lipids such as free cholesterol, cholesteryl esters, triacyl- and diacyl-glycerols [19,20]. Bioactive lysosphingolipids such as sphingosine-1-phosphate (S1P), sphingosylphosphorylcholine (SPC) and lysosulfatide (LSF) play an important role in HDL signalling; other cargo molecules include hormones, carotenoids, vitamins and microRNA sequences (Figure 1) [21,22,23]. The HDL microRNA cargo is discussed in detail in Section 2.2 and Section 3.

### 1.2. Assembly, Maturation and Uptake of HDL Particles

The most abundant apolipoprotein (apo) component of HDL is apoA-I, a 28kDa protein synthesized in the liver and the intestine [1,2,3,24]. Formation of HDL is dependent upon the initial lipidation of apoA-I via the ATP-binding cassette transporter A1 (ABCA1), the gatekeeper of the reverse cholesterol transport pathway [1,2,3,24]. Dimers of apoA-I accept phospholipids and free cholesterol from ABCA1 transporters to form nascent (discoidal) HDL particles, which can then acquire additional lipids via ABCG1, and undergo maturation in the blood stream. This involves activation of lecithin–cholesterol acyl transferase (LCAT), that generates hydrophobic cholesteryl esters forming the core of spherical, lipid-rich HDL particles, which mature further via the function of the cholesteryl ester transfer protein (CETP) and phospholipid transfer protein (PLTP) [1,2,3,24].

Scavenger receptor B1 (SR-B1) can also efflux cholesterol to mature HDL particles via passive diffusion in a manner which is concentration dependent, but the major role of this protein appears to be the delivery of cholesterol and cholesteryl ester from HDL to the liver, for excretion in bile in the form of cholesterol or biliary lipids [25]. SR-B1 mediates the non-endocytic, selective uptake of HDL-cholesteryl esters (and cholesterol) and is generally recognised as the physiologically functional HDL receptor [25]. 

### 1.3. HDL: Protective Biological Functions

High-density lipoproteins, and their apolipoproteins, transporters and receptors, bring about a plethora of effects within the innate and adaptive immune responses. The canonical pathway by which HDL and HDL apolipoproteins exert protective functions is via removal of excess cholesterol from peripheral tissues, via interaction with ABCA1, ABCG1 and ABCG4 and SR-B1 (above). For example, HDL prevent the accumulation of cholesterol and cholesterol crystals within cells and tissues, avoiding activation of the NLRP3 (nucleotide-binding domain (NOD)-like receptor protein 3) inflammasome [26]. The efflux of cholesterol from macrophages via ABCA1 and ABCG1 transporters limits the cholesterol content of the plasma membrane available for lipid raft formation, inhibiting MyD88 (myeloid differentiation primary response 88) and Toll-like receptor (TLR) trafficking and signalling and nuclear factor kappa B (NF-κB) activation [27,28] as reviewed in [29]. Based on the CANTOS (Canakinumab Antiinflammatory Thrombosis Outcome Study) trial, Westerterp et al. (2018) hypothesized that interleukin 1-β dependent NLRP3 inflammasome activation is linked to cholesterol transporters ABCA1 and ABCG1 [30]. A deficiency of Abca1/g1 in myeloid cells increased activation of the NLRP3 inflammasome in LDL receptor knockout mice, while deficiency of NLRP3 was able to decrease atherogenesis in mice genetically deficient in Abca1/g1 [27,28] (Figure 2). Both transporters are associated with the HDL-mediated suppression of haematopoetic stem cell proliferation [28,29,30]. The loss of Abca1/g1 markedly increased neutrophil recruitment and extracellular trap (NETosis) formation in lesions, and patients with a loss of function mutations in ABCA1 (Tangier disease) exhibit increased myeloid cholesterol content and inflammasome activation [28,29,30].

Cholesterol is synthesized endogenously in and around the endoplasmic reticulum (ER), or derived from endocytosed low-density lipoprotein (LDL) or oxidized LDL (OxLDL), processed in the late endosome/lysosomes by lysosomal acid lipase (LAL). Free cholesterol (FC) can be stored as lipid droplets of cholesteryl ester (CE) after esterification in the ER. Cholesterol can be removed from the cell (efflux) by trafficking from the ER, late endosomes or hydrolysed lipid droplets, and transported to the plasma membrane for transfer to lipid-poor apolipoprotein (ApoA)-I or nascent HDL via ATP binding cassette transporters (ABC) transporters (ABCA1, ABCG1). Scavenger receptor B1 (SR-B1) mediates both influx and efflux of cholesterol from HDL, in a concentration-dependent manner. Via these routes, HDL prevents the accumulation of FC, CE and cholesterol crystals within cells, limiting the activation of the NLRP3 (nucleotide-binding domain (NOD)-like receptor protein 3) inflammasome, and reducing the release of pro-inflammatory cytokines such as interleukin (IL) IL-1β, IL-18, IL-6 and tumour necrosis factor (TNF)-α. Efflux of cholesterol also inhibits MyD88 (myeloid differentiation primary response 88) and Toll-like receptor 4 (TLR4) signalling, via IRAK (IL-1 receptor associated kinase)-1 and -4 and TRAF6 (TNF receptor-associated factor 6), reducing nuclear factor κB (NF-κB) activation.

The ABCA1 and ABCG1 transporters also play important roles in the regulation of adaptive immune responses mediated by T-lymphocytes. T-cell receptor responses are strongly influenced by the composition and lipid raft structure of the plasma membrane (reviewed in [31]). T-cell activation decreases expression of ABCA1 and ABCG1, which is associated with the activation of protein tyrosine kinases Lck (lymphocyte cell-specific protein-tyrosine kinase) and Zap 70 (zeta-chain-associated protein kinase 70), and the activation of MAP (mitogen-activated protein kinase) and ERK (extracellular signal-regulated kinase) 1/2 kinases [32]. The genetic deletion of ABCG1 increases the proliferation of thymocytes during their transition phase, resulting in increased numbers of CD (cluster of differentiation) 4+ single positive thymocytes. Equally, the lack of this transporter-enhanced T-cell receptor signalling increases Zap70 and Erk1/2 phosphorylation [32]. By contrast, induction of ABCG1 expression via Liver X receptor (LXR) signalling reduced T-cell proliferation [33]. However, loss of ABCG1 exerts opposing effects on natural killer T-cells (NKT), impairing their maturation and proliferation [34]. Cholesterol homeostasis via ABCG1 is also central to the role of regulatory T-cells (Tregs), which suppress innate and adaptive immune responses [35]. In the absence of the ABCG1 transporter, there is a selective expansion of Tregs [35] that reflects the role of elevated cholesterol. In turn, this inhibits mammalian target of rapamycin (mTOR) signalling, and favours STAT5 (signal transducer and activator of transcription) activation. Lipid rafts and membrane cholesterol content also promote B-lymphocyte receptor signalling and endocytosis [36,37].

Notably, ABCA1 can also act as a tumour suppressor, promoting cell death and limiting AKT (protein kinase B) signalling via regulating the cholesterol content of the plasma membrane; missense mutations in ABCA1 are associated with enhanced proliferative potential in patients with chronic myelomonocytic leukaemia [38]. High-density lipoproteins [39,40,41,42], apoA-I [41,43,44,45] and ATP-binding cassette transporters [46,47,48,49,50] also protect beta cells and pancreatic islets in both experimental models and clinical studies [51,52,53,54,55]. While some of these protective functions have been ascribed to sterol removal from beta cells [39,43,44,45,46,47], other reports cite the cholesterol- and/or transporter-independent effects of apoA-I and HDL [41,44,56].

Incubations with HDL and apoA-I reduce the adherence of polymorphonuclear (PMN) leucocytes to endothelial cells, through blocking the activity of lipopolysaccharide (LPS) and modifying the expression of CD11b/CD18 on PMN via interaction with SR-B1 and ABCA1, respectively [29]. Further, SR-B1 aids the conversion of macrophages from the inflammatory M1 to the anti-inflammatory M2 phenotype, and reduces NF-κB, p38 and Janus kinase (JNK) signalling [57]; mice lacking SR-B1 show enhanced expression of inflammatory cytokines, and a genetic variant of this receptor has been associated with increased risk of coronary heart disease [58]. Macrophage SR-B1 is implicated in autophagy [59], and can mediate the efferocytosis or removal of apoptotic cells, promoting the survival of phagocytosis and an anti-inflammatory response [25]. This occurs via the Src (proto-oncogene tyrosine protein kinase Src)/PI3K (phosphoinositide-3-kinase)/Akt signalling pathway; recent data has demonstrated a direct interaction between the PDZK1 domain of SR-B1 and phosphorylated Src, which promotes endothelial nitric oxide synthase (eNOS) signalling and enhances endothelial cell growth and migration [25]. Equally, myeloid-derived suppressor cells (MDSCs), which potently inhibit adaptive immune responses, can be targeted by HDL interactions with SR-B1 to deliver a possible anti-tumour immune response. Treatment with HDL-like nanoparticles blocks MDSC in vitro and in vivo, reducing tumour size and metastatic tumour burden [60]. HDL also promotes angiogenesis [61,62,63,64] and wound healing [64,65,66] through the interaction of S1P with S1P3 receptor [61] and via interactions with SR-B1 and activation of the PI3K/Akt pathway. This increases the proliferation and migration of endothelial cells, and promotes re-epithelialization. The interaction of HDL with SR-B1 (above) and the apolipoprotein E receptor (apoER2/LDL receptor-related protein 8) also mediate the inhibition of agonist-dependent aggregation of platelets, binding of fibrinogen, granule secretion and the release of thromboxane A_2_ (TxA_2_) [67].

The impact of HDL on inflammatory responses may also reflect the ability of this lipoprotein to limit oxidative events and their sequelae. These properties are ascribed to the apolipoprotein content, and the presence of enzymes such as paraoxonases (PON1, PON3) and PAF acetylhydrolase. HDL can reduce intracellular oxidative stress, limit oxidation of low-density lipoprotein (OxLDL) and OxLDL-induced apoptosis, and prevent activation of inflammatory NF-κB signalling, thereby limiting the expression of adhesion molecules and chemokines (reviewed in [68,69]), further supporting endothelial physiology.

The mechanisms by which HDL and its cargo mediate these multiple complex events involve various interactions with G protein-coupled receptors, ABC transporters, SR-B1, ectopic β2-ATPase and purinergic ADP receptors (P2Y_12/13_), Hedgehog (hH), Smoothened (SMO) and S1P_1-5_ receptors [42], as is reviewed in [70]. In turn, HDL triggers diverse signalling pathways [71]. These include: protein kinases A, B and C, forkhead box protein 01 (Fox01), MAPK, nitric oxide synthase, c-Jun N-terminal kinase (JNK), cell division control protein 42 (Cdc42), Janus kinase 2 (JAK2), STAT3, liver kinase B1 (LKB1), calcium calmodulin-dependent protein kinase (CAMK) and AMP kinase (AMPK) among others, eliciting changes in gene and protein expression [70]. These last also lie under the control of epigenetic mechanisms, including microRNA sequences that can be modified by exposure to HDL reviewed in [71], and discussed below.

## 2. MicroRNA 

### 2.1. MicroRNA Biogenesis and Function 

The impact of miRNA on HDL biogenesis and metabolism is well established and has been covered in several comprehensive review articles [71,72,73]. Instead, this review will focus on how HDL can change the cellular miRNA landscape, and seek to identify differentially expressed sequences that relate to the protective mechanisms attributed to this lipoprotein.

MicroRNA are small (~19–25 nucleotide) non-coding RNA sequences which modify the expression of gene networks, found isolated or clustered within the human genome, and transcribed from intergenic or intronic regions [74,75] by RNA polymerase II/III [76,77]. Transcription can occur via both canonical and non-canonical pathways [78,79,80]; in the former, the formation of a primary miRNA (pri-miRNA) transcript—containing a hairpin loop, a 5′-methylated cap and a 3′-polyadenylated tail—is processed to precursor (pre) microRNA by a complex containing the RNA-binding protein DiGeorge syndrome critical region 8 (DCGR8) and the ribonuclease III Drosha [81,82]. Pre-miRNA, around 70 nucleotides in length, are exported from the nucleus by exportin-5 and RanGTP, and processed in the cytosol by Dicer (RNase III) to the mature miRNA duplex (19–25 nucleotides) [83,84,85,86]. The guide strand is loaded onto the active RNA-induced silencing complex (RISC), composed of Dicer, TAR RNA-binding protein (TRBP) and Argonaute (1-4) proteins, and the miRNA recognition element guides the base pairing with complementary mRNA molecules [87,88,89,90].

Perfect or near-perfect complementary matches between miRNA and conserved 3′-UTR regions of target mRNA degrades the mRNA; imperfect complementarity results in moderate reductions in mRNA levels, and translational repression [91,92,93,94,95]. Thus, the outcome is reduced protein expression from the gene targeted, which can either be quite modest or more pronounced in extent. MicroRNA are therefore ‘fine-tuning’ gene and protein expression, and thereby modifying cell function [91,92,93,94,95]. However, each miRNA sequence can target hundreds of genes within networks: it has been suggested that more than 60% of all genes encoding proteins in the human genome are miRNA targets [92,93,95,96]. Notably, miRNA sequences are expressed in a tissue-specific manner, and concentration-dependent effects occur, particularly in diseased compared to healthy tissues [97,98,99,100]. MicroRNA sequences, including those circulating in the bloodstream, can be modified in health and disease, including ageing and ageing-related diseases reviewed in [101] and traits associated with the metabolic syndrome reviewed in [102]. These sequences can prove to be valuable biomarkers of disease and disease progression [101,102,103,104,105,106], some of which have also been linked to changes in HDL-C levels in the bloodstream [107].

### 2.2. HDL microRNA Cargo: Cellular Export and Import of microRNA to HDL

Perhaps the most obvious way in which HDL may alter the cellular miRNA landscape is via the direct export or import of these HDL cargo sequences by cells and tissues. While some miRNA sequences are transported in the bloodstream by HDL and other lipoprotein particles [107], others are transported in extracellular vesicles—such as exosomes [108], microvesicles [109] and apoptotic bodies [110], and by ribonucleoproteins such as Argonaute 1 and 2 (AGO1/2) [111,112]—significantly increasing their stability compared to naked microRNA. Extracellular vesicles carrying microRNA can be generated from distinct intracellular sites: microvesicles are thought to be generated by budding of the plasma membrane, while exosomes are released from the endosomal compartment, upon endosome budding, followed by the fusion of multi vesicular bodies (MVBs) in the plasma membrane reviewed in [113]. At present, over 20 endogenous microRNA sequences (around <1% of the total microRNA landscape) have been identified in HDL [107], with miR-24, miR-33, miR-92a, miR-146a, miR-223 and miR-486 being among the most abundant, and subject to regulation by pathophysiological conditions such as diabetes, infection, cancer, infection and inflammation [21,107]. Although characterisation of the HDL-miR cargo is a challenging proposition, Seneshaw et al. (2016) [114] have developed a high-throughput methodology for identification of miRNAs in HDL from plasma samples, while Ishikawa et al. (2017) [115] have demonstrated the significant stability of serum HDL microRNA under differing storage conditions, including treatment with RNase A and multiple freeze–thaw cycles.

The mechanism through which export to HDL is achieved is not fully characterized, although macrophages are reported to export miRs to HDL in a manner that is inversely related to exosome release from the secretory pathway, and independent of SR-B1 [116,117]. The generation of ceramides by neutral sphingomyelinases can also facilitate export to exosomes and HDL, presumably via functionally distinct and possibly opposing pathways [21]. Desgagne et al. (2019) demonstrated that the microtranscriptome of HDL differs substantively from that of plasma [117]; notably, a common determining sequence for the export of miRs to HDL has not been reported, although it is clear that this is a selective process, as exported miRNA sequences do not simply reflect the relative abundance of those sequences within cells [21,23]. Most recently, Michell et al. (2022) explored the physio-chemical factors influencing the interaction of HDL with small RNA, finding a critical role in the non-ionic interactions with apoA-I [118].

Export also seems to be by cell-type and/or specific: in human pancreatic beta cells, export of miR-375 to HDL is inversely linked with insulin secretion, and regulated by high glucose [119], while miR-122-5p is thought to be hepatocyte-specific [23] and miR-223-3p to be derived from macrophages [116]. Neurons transfer miR-124-5p and miR-9-5p to HDL via a mechanism dependent upon electrical signalling, and HDL can transport miR-124-5p from neurons to microglia [120]. There are reports that microRNA sequences can be transferred between cells via gap junction (GJ) connexins, providing a route for the coordinated regulation of gene expression within tissues, with the GJ transfer of miR-34a inhibiting the proliferation of glioma cells [121]; HDL can also regulate formation of connexin 43 GJ, potentially modifying these responses [122].

Despite the predominant role of SR-B1 as the physiological HDL receptor (above), it not clear whether this receptor is responsible for uptake of HDL microRNA cargo in all cell types; indeed, Wolfrum et al. (2007) described a role for the interaction between HDL-apoE and the LDL receptor in the delivery of siRNA [123]. While SR-B1 is reported to be responsible for HDL-miRNA delivery in human hepatocytes [21] and porcine endothelial cells [124], the global deficiency of SR-B1 in mice did not cause substantive differences in small RNA levels in HDL, compared to wild-type mice, suggesting that this receptor may not be a critical regulator of HDL-miRs in vivo [125]. Importantly, this study, which employed small RNA sequencing, revealed that most small RNA sequences carried by HDL and other lipoproteins are not endogenously derived, but from bacterial sources in the microbiome and the environment [125]. At present, it remains unclear whether the interaction of HDL with cell signalling receptors (above) can influence microRNA uptake and further work on the uptake of HDL microRNA cargo, whether exogenously or endogenously derived, is clearly warranted [21,125].

### 2.3. Regulation of Endogenous microRNA Expression by HDL

Notably, HDL can alter microRNA sequences in cells and tissues without direct export or delivery, but by modifying their endogenous expression, and these processes undoubtedly intersect. Regulation of the expression of microRNA is highly complex, and occurs at every stage of their production, from transcription via RNA polymerase [74,75,76,77], through multiple processing stages and modulation of stability, to both export and import reviewed in [126,127]. In mammalian cells, around 70% of miRNAs are located within introns, the majority within protein-encoding genes; the remainder are found in non-coding RNA transcription units (intronic or exonic) [126]. Host gene and miRNA expression can occur in parallel, reflecting related functions: for example, intronic miR-33 located within the gene (*SREBF2*) encoding sterol regulatory element binding protein 2 (SREBP2), a transcriptional regulator of cholesterol synthesis, inhibits the expression of ABCA1, thereby limiting cholesterol efflux to apoAI, and aiding the coordination of cholesterol homeostasis [128,129]. Alternatively, the expression of the host gene and miRNA can be uncoupled via use of alternative promoters and/or regulation of miRNA processing: about one third of intronic miRNAs are transcribed independently of their host gene, with additional start sites which can be thousands of nucleotides upstream of the miRNA sequence [126].

A significant number of transcription factors have been identified that enhance or repress the synthesis of pri-miRNAs: in many cases, these transcription factors are themselves targets of those miRNA sequences, providing feedback loops that can amplify or modulate their own expression [130]. Their expression can also confer tissue specificity: serum response factor (SRF), MyoD, Mef2 and myogenin regulate the expression of miR-1 and miR-133a, and are restricted to muscle, while transcription factors (forkhead box P3 (Foxp3), NF-κB and activator protein 1 (AP-1)) that regulate miR-155 are critical regulators of the immune system [131,132,133]. However, there are a host of regulators of miRNA biogenesis, including DEAD-box (DDX) RNA helicases (p68/DDX5; P72/DDX17) which facilitate the processing of pri-miRNA, and Suppressor of Mothers against Decapentaplegic (SMAD) transcription factors which not only regulate transcription, but promote pri-miRNA processing as reviewed in [126]. DNA- and RNA-binding proteins can either stimulate or repress the processing of pri- or pre-miRNA [126]. These factors include the transactive response DNA-binding protein 43 kDa (TDP-43), breast cancer susceptibility gene A1 (BRCA1), KH-type splicing regulatory protein (KSRP), Lin-28 homolog A (LIN-28), heterogeneous nuclear ribonucleoprotein A1 (hnRP A1), transaction response element RNA-binding protein (TRBP) and RNA-binding protein 3 (RMB3)) [126]. Similarly, the stability of miRNA sequences can be modified by RNA-binding proteins such as Argonaute (above) and QK1 (Quaking homolog, KH domain RNA binding 1), the cytoplasmic poly(A) polymerase GLD-2 and the 5′-3′ exonuclease XRN-1/2 [126].

So, can HDL treatment be linked with modified cellular expression or the activity of any of the factors involved in the generation and regulation of microRNA biogenesis? Correlations between the gene expression of Drosha and apoA-I, and levels of pri-miR-223 and miR-223-3p, occur in plaque derived from patients with carotid artery stenosis (CAS) and associated with dysfunctional HDL [134]. Hyperglycaemia also increases Drosha, DGCR8 and Dicer in macrophages, increasing levels of miR-233, miR-92a and miR-486 in HDL in patients with acute coronary syndrome (ACS) [135]. Notably, Torres et al. (2015) demonstrated that exposure of neutrophils or macrophages to HDL increased the gene expression of Dicer, and variously enhanced the biogenesis and/or export of miR-223 in these cells [116]. Heterozygous knock-in of a mutated form (TDP-43(A315TKi) of the DNA-binding protein, TDP-43, which stimulates pri- or pre-miRNA processing [126,136], is associated with increased levels of HDL-cholesterol in the bloodstream [137]; similarly, HDL-C has been linked with myogenin and myoD expression [138]. By contrast, many studies report that native and reconstituted HDL can directly repress the activation of NF-κB in a variety of disease contexts, providing one route to the regulation of miR biogenesis [126,139,140,141,142,143,144,145]. This lipoprotein fraction can also limit endothelial to mesenchymal transition (EMT), induced by transforming growth factor–β (TGF-β), via induction of inhibitory SMAD7 [146]. Lin et al. (2021) recently demonstrated that the degree of HDL functionality can be related to modulation of SMAD2/3 and Snail during EMT [147], while ApoA-I can promote cardiogenic maturation from embryonic stem cells via the bone morphogenetic-4 (BMP-4)-SMAD signalling cascade [148,149].

Cell signalling pathways activated by HDL [70] can modify microRNA biogenesis, and *vice versa*. However, these pathways are activated by multiple factors, including cytokines, hormones and growth factors. For example, pre-implantation factor (PIF) activates PKA and PKC, and also inhibits let-7 production [150] while HDL interaction with SR-B1 induces both miR-223-3p production and the activation of protein kinase C [116], but direct causality between these events has not been established. The hypoxia-induced transcription factor Nur77 suppresses the expression of Dicer and let-7i-5p, resulting in the activation of PI3K/Akt [151]. By contrast, aerobic exercise training increases the expression of Dicer in both muscle and adipocytes, via activation of AMPK, leading to an overall increase in miRNA sequences, including miR-203-3p, which limits glycolysis in adipose tissue under conditions of metabolic stress [152].

Perhaps one of the most established routes by which HDL may directly modulate microRNA biogenesis is via regulation of the activity of the transcription factor STAT3. For example, HDL and its constituent sphingosine-1-phosphate (S-1-P) increase the tyrosine and serine phosphorylation of STAT3 in ventricular cardiomyocytes via the S1P2 receptor and activation of phospholipase C-β, ROCK (Rho-associated protein kinase), ERK1/2 and Src [153,154,155]. Notably, Pedretti et al. (2019) found that HDL mediates STAT3-dependent repression of miR-34b and miR-337 expression [154]; however, signalling via S-1-P is cell-type specific, as its molecular action depends upon the differential expression of its receptors (S1PR1-S1PR5) in differing tissues [156]. Certainly, a complex relationship exists between STAT3 activation and miR regulation, with miRs regulating STAT3 expression and vice versa as reviewed in [157,158]. The induction of miR-21 via STAT3 is noted in multiple myeloma cells [159], while STAT3-mediated suppression of miR-199a-5p occurs in cardiomyocytes and endothelial cells [160].

## 3. Modification of the Functional Cellular microRNA Landscape by HDL

Treatment with HDL can therefore alter the cellular miRNA landscape via the delivery, export or modulation of the endogenous expression of miRNA sequences (Figure 3). Arguably, knowledge of the miRNA sequences, and their target gene networks, responsive to HDL treatment, or susceptible to dysfunctional HDL, could provide access to therapeutic strategies for complex disease entities. This is a key point: harnessing the therapeutic potential of HDL has proved challenging in many respects, in part due to the heterogeneous and relatively unstable nature of this lipoprotein and its variable cargo molecules [11,12], and HDL dysfunction is a key contributing factor in a number of disease processes [15,16,17,18] as reviewed in [161,162,163,164,165,166].

Interrogation of the literature (NCBI PubMed 21/07/2023: ‘HDL’ and ‘microRNA’) generates 408 outputs, the majority of which focus on the role of microRNA in the regulation of the expression of genes and proteins involved in cholesterol efflux and homeostasis, such as miR-33a/b [128,129] and miR-223 [22]. More recently, Lu et al. (2022) demonstrated that miR-320b is highly expressed in peripheral blood mononuclear cells from individuals with coronary artery disease compared to healthy controls [167]. This sequence decreased cholesterol efflux to HDL and apoA-I through targeting ABCG1 and EEPD1 (endonuclease-exonuclease-phosphate family domain 1) and repressing the LXRα/ABCA1/G1 pathway [167]. ApoE^−/−^ mice exposed to this sequence responded in the same way, with increasing atheroma and lesional macrophage content, production of inflammatory cytokines and activation of NF-κB [167].

The first reports that apoA-I and HDL deliver siRNA [123,167,168,169] and microRNA [21] that can directly modify the functions of cells and tissues, came from Wolfrum et al. (2007) [123], Lee et al. (2009) [168,169], Kuwahara et al. (2011) [170] and Vickers et al. (2011) [21]. Notably, in their ground-breaking study, Vickers et al. (2011) [21] prepared highly purified HDL from human plasma using density gradient ultracentrifugation, followed by fast-protein liquid chromatography (FPLC) and anti-apoA-I immunoprecipitation, which were negative for exosomal protein markers. Significant changes in abundance were noted for sequences carried by HDL from healthy individuals compared to that isolated from patients with familial hypercholesterolaemia (↓miR-632, miR-625, miR-573, miR-572, miR-509-3p, miR-323-3p, miR-520c-3p, miR-877, miR-138-1, miR-135a; ↑miR-24, miR-188-5p, miR-412, miR-191, miR-218, miR-320, miR-222, miR-223, miR-126, miR-323-3p, miR-106a, let-7b). Significantly, both exogenous and endogenous miRs delivered to cells by HDL via SR-B1, proved capable of directly targeting mRNA reporters, inducing differential gene expression and promoting loss of established gene targets in cultured (Huh7) hepatocytes [21]. Moreover, reconstituted HDL, free of any RNA sequences, were able to retrieve miRs from the bloodstream when injected intravenously into wild-type or apoE^−/−^ mice, with the profiles of the resulting sequences differing substantively in atherosclerotic mice compared to controls [21].

The same group went on to demonstrate that the transfer of miR-223 from HDL to endothelial cells (which do not transcribe this sequence) repressed translation of one of its direct targets, Intercellular Cell Adhesion Molecule-1 (ICAM-1) [171]. Thus, at least part of the anti-inflammatory properties of HDL may be ascribed to the delivery of miR-223, which is a key regulator of immune cell differentiation and inflammation (reviewed in [172]). Notably, however, miR-223 can also be delivered by extracellular vesicles, and is endogenously regulated (positively and negatively) by transcription factors such as C/EBP α/β (CAAT/enhancer binding protein α/β), PU.1, nuclear factor IA and Kruppel–like factor 4 (KLF4) [172]. MiR-223 is also part of a feedback mechanism contributing directly to the inhibition of cholesterol biosynthesis (via repression of HMG CoA synthase 1 and methylsterol monooxygenase-1) and the uptake of HDL (SR-B1), and indirect promotion of the expression of ABCA1 via transcription factor Sp3, enhancing cholesterol efflux [22].

Not all reports confirm the functional outcomes, or delivery, of microRNA from HDL to cells: For example, Wagner et al. (2013) [173] analysed HDL from healthy individuals and from patients with stable CAD, or acute coronary syndrome. This group also found the most prominent sequence to be miR-223 (detected at >10,000 copies μg HDL^−1^ protein), but levels of this sequence did not vary markedly between the groups tested, and these authors did not find substantive delivery to endothelial, smooth muscle or peripheral blood mononuclear cells [173]. Axman et al. (2018) also indicated low uptake of HDL-miRs by Chinese Hamster Ovary cells (CHOK1) and receptor-modified derivatives, concluding that any influence on cell metabolism via lipoprotein uptake was highly unlikely, although they could demonstrate the uptake of HDL artificially loaded with microRNA sequences [174].

By contrast, Riedel et al. (2015) found that the ability of HDL to stimulate nitric oxide generation is diminished in lipoproteins isolated from patients with coronary heart disease, in which levels of the pro-angiogenic sequences miR-126 and miR-21 were reduced (a finding that could be attenuated by the exercise training in those individuals) [175]. Conversely, marked increases in anti-angiogenic HDL-bound miR-181c-5p were observed in Australian Aboriginal people with diabetic macrovascular complications, compared with healthy controls or individuals with type 2 diabetes from the same indigenous group, a finding associated with reduced angiogenic capacity as judged using tubule formation in human coronary aortic endothelial cells [176]. Notably, transcoronary concentration gradients for HDL-miRs [100,177], with greater depletion of miR-16, miR-92a and miR-223, were noted in patients with acute coronary syndrome, compared to patients with stable coronary artery disease; these changes were not associated with changes in HDL composition or size [177].

The concept of the remodelling of the HDL microRNA cargo in disease contexts was initially established by Vickers et al. (2011) [21] and above [176,177]. Diet-induced changes have also been observed: a high-protein diet (30% of energy intake; 12 weeks) induced significant weight loss, correlated with decreases in HDL-miR-223, in overweight and obese men [178]. More recently, Ben-Aicha et al. (2020) explored the impact of a high fat/cholesterol diet on HDL-miRNA and recipient porcine aortic endothelial cells [124]. In brief, HDL-miRNA profiling of 149 miRs revealed four differentially regulated sequences in HC-HDL compared with normocholesterolaemic (NC)-HDL, with higher levels of miR-126-5p/3p and miR-20-5p noted in HC-HDL, and lower levels of miR-103-3p and miR-let-7g-5p [124]. Incubation of aortic endothelial cells with HC-HDL (but not NC-HDL) caused a SR-B1 dependent 2.5-fold increase in miR-126 (and, intriguingly, *only* miR-126) in endothelial cells, associated with marked reduction in its key target, hypoxia inducible factor (HIF)-1α [124]. An interesting commentary highlights the lack of information on the cellular mechanisms governing the specificity of miRNA uptake from HDL, and suggests that HDL-miR cargo could prove ‘a lasting memory of a high fat diet’, and possibly an indicator of the (potentially reversible) atherogenicity of this lipoprotein fraction [179].

HDL-like nanoparticles have also been specifically tailored to deliver SiRNA (individual or duplex pairs) and miRNA via SR-B1 to regulate gene expression [123,168,169,180,181,182,183], and can achieve anti-tumour results [181,182,183]. Wang et al. (2020) delivered miR-205 to primary human corneal epithelial cells, wherein topical administration of nanoparticles containing miR-205 improved wound closure, compared to HDL carrying a scrambled miR, when applied to corneal epithelial debridement wounds, and significantly aided resolution of inflammation [180]. However, it should be noted that HDL nanoparticles themselves also improved wound closure, reinforcing the myriad functions associated with this type of particle (above) [180]. HDL nanoparticles have also been used to deliver radio-sensitising miR-34a in head and neck cancer cell models, resulting in reduced metabolic activity and increased apoptosis following radiation treatment [183].

Certainly, the miRNA sequences altered by cellular interactions with HDL can differ from those induced by exposure to isolated and purified apoA-I [184], which does not contain S-1-P, including sequences which are not carried in the HDL particle. While it is clear that multiple microRNA sequences modulate cellular cholesterol efflux and regulate the expression of ABCA1/G1 [22,71,72,73,128,129,167,185,186], the changes in the miRNA profile induced by HDL are less well characterized; it is also unclear which sequences are modified by cholesterol efflux, or by the myriad other properties associated with this lipoprotein. In human PANC 1.1B4 hybrid pancreatic cells, wherein both HDL and apoA-I induce cholesterol efflux to differing extents, the sequences regulated by exposure to apoA-I and HDL include elevations of hsa-miR-25-3p and hsa-miR-29a-3p and reductions in hsa-miR-30e-5p, hsa-miR-146a-50 and hsa-miR-335-3p, as judged using microchip array analysis [184]. However, only HDL increased the expression of hsa-miR-21-5p and hsa-miR-7977, and reduced levels of hsa-miR-16-5p, hsa-miR-130a-3p and hsa-miR-191-5p, while a miR-21-5p mimic replicated HDL repression of *SMAD7* and *STAT3* in 1.1B4 cells [184]. Notably, except for hsa-miR-130a-3p and hsa-miR-7977, the other sequences regulated by HDL are found in serum or plasma, but are not reported in HDL (DIANA PlasmiR, accessed 20 July 2023) [187].

Reconstituted HDL (rHDL) variously inhibits the expression of anti-angiogenic hsa-miR-181c-5p in human cardiac microvascular endothelial cells, under normoxic, hypoxic and inflammatory conditions [188]. While mmu-miR-181c-5p does not occur in plasma or HDL, rHDL suppresses the expression of this sequence, and rescues hindlimb angiogenesis during early post-ischaemia in diabetic mice, suggesting that rHDL targets mmu-miR-181c-5p specifically at the ischaemic site [188]. The mechanism by which this occurs remains unknown, although tissue-specific miR expression is an established concept [97,98,99,100]. However, it is clear that native HDL from healthy subjects can promote angiogenesis, while that derived from patients with coronary artery disease or diabetes attenuates this response [61,62,63,64,65]; native HDL suppresses miR-24-3p, and enhances vinculin expression, resulting in increased production of nitric oxide [188,189]. In contrast, dysfunctional HDL derived from CAD patients delivered miR-24-3p via SR-B1, inhibiting vinculin expression and NO production, and leading to superoxide production, while the overexpression of vinculin or inhibition of miR-24-3p reversed the impaired angiogenesis associated with dysfunctional HDL [189].

The impact of HDL on the cellular miR landscape also seems cell-maturation dependent, at least in human dermal fibroblasts (HDF) [190]. In primary neonatal(n) and adult(a) human dermal fibroblasts, exposure to HDL increases cholesterol efflux, cellular viability and ‘scrape’ wound healing, and multiple changes in expression of miR sequences, as judged using microchip array [190]. Expression of hsa-miR-6727 was significantly increased in HDFn and decreased in HDFa [190], while a hsa-miR-6727 mimic repressed different target genes in HDFn (*ZNF584*) and HDFa (*EDEM3*, *KRAS*). The mimic promoted wound closure in HDFn cells, while the hsa-miR-6727 inhibitor promoted wound closure in HDFa fibroblasts. Thus, it is clear that one of the sequences modified by exposure to HDL can influence wound closure in both neonatal and adult fibroblasts, but cannot replicate the myriad effects mediated by exposure to HDL [190].

Thus, exploitation of the changes in the cellular microRNA landscape associated with exposure to HDL may provide an alternate therapeutic strategy to treat disorders such as inflammation and immunity, wound healing, angiogenesis, dyslipidaemia, atherosclerosis and coronary heart disease. To date, clinical translation of HDL-targeted therapies has not been fully realised; the development of CETP inhibitors has proved challenging with three compounds failing in phase III clinical trials [1,191]. Although the REVEAL trial demonstrated that anacetrapib decreased CHD in combination with statin therapy, these effects seem to be due to a lowering of non-HDL cholesterol, rather than raising HDL-C as reviewed in [191]. The AEGIS-II phase III trial is currently studying the potential benefits of a novel formulation of apoA-I (CSL112), for treatment of the high-risk period (90d) after acute myocardial infarction; CSL112 is designed to promote cholesterol efflux from macrophages within atherosclerotic plaque, and reduce inflammation [192]. The treatment strategies targeting HDL have largely focused on cardiovascular disease [1,191,192], but these drugs may exert protective outcomes in other clinical contexts.

Over the last decade or so, the concept of RNA-based therapeutics, including small non-coding RNA sequences, has gained considerable traction [193,194,195,196,197,198,199]. As Winkle et al. (2021) point out, the use of miRNA therapeutics may be advantageous: miRs are naturally occurring molecules, unlike antisense oligonucleotides, such that endogenous mechanisms are in place for their processing and target selection [198]. They can also target multiple genes within a specified pathway, providing a more effective response [198]. If existing problems such as delivery, immunogenicity and potential off-target effects can be eliminated, a wealth of potential therapeutic indications may become realised.

## 4. Conclusions

In conclusion, it seems evident that HDL can achieve epigenetic regulation via altering the cellular microRNA landscape, via multiple mechanisms including direct delivery, but also possibly via initiation of cholesterol efflux, modulation of inflammation and cell signalling. These modalities demand further investigation, as do the roles of microRNA sequences derived from exogenous sources such as bacteria and the environment. Regulation of the mechanisms governing microRNA uptake and secretion remain a key area for investigation, as does the relationship between HDL (dys)function and the promotion of disease via modulation of microRNA sequences, offering new routes for therapeutic intervention.

## Figures and Tables

**Figure 1 biology-12-01232-f001:**
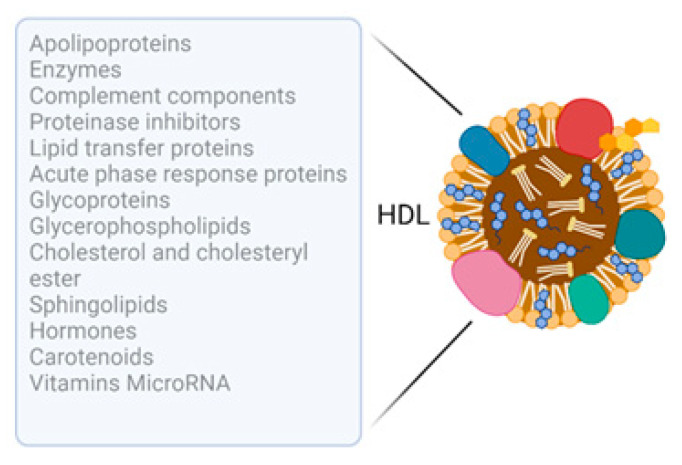
Schematic summarising the cargo molecules carried by high-density lipoproteins.

**Figure 2 biology-12-01232-f002:**
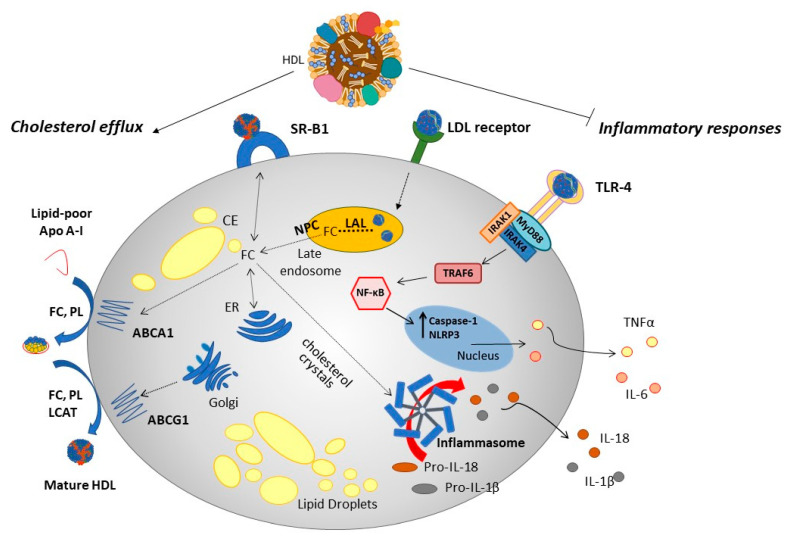
Cellular functions influenced by HDL.

**Figure 3 biology-12-01232-f003:**
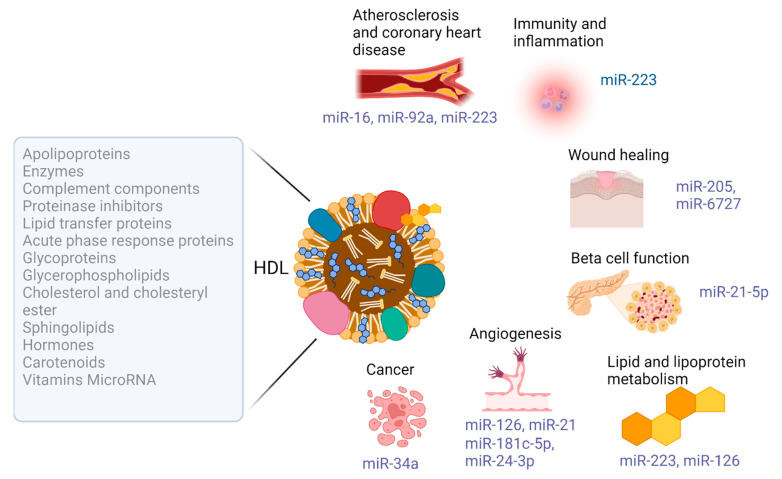
A summary of the text, indicating the components of high-density lipoprotein, and the impact of this lipoprotein on the microRNA profile in cells and tissues, in differing disease contexts.

## Data Availability

This review article draws on material freely available from PubMed, NCBI.

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
