# Peer review of "Modulation of the Cellular microRNA Landscape: Contribution to the Protective Effects of High-Density Lipoproteins (HDL)"

_biology, 2023, doi:10.3390/biology12091232_

Round 1
Reviewer 1 Report
This is a nice and very detailed review on HDL with a focus on induction and delivery of miRNA with implications on various cellular processes.
The review is very nicely written and covers all the relevant contributios/papers.
Minors:
lane 98: B-1 should be B1
lane 196: fo S1P should be of S1P
lanes 268-269: it not clear this receptor - it is not clear whether/if
this receptor
lane 350: whichlimits - which limits
Very nice review!
Author Response
I thank the reviewer for their kind words, and for their time taken in careful reading of this manuscript. My responses are highlighted within the manuscript for convenience of checking.
- Line 98: ‘B-1’ is now changed to ‘B1’
- Line 196 (170): ‘fo’ S1P now changed to ‘of’ S1P
- Lines 268-269: ‘it not clear this receptor’ is now changed to ‘it is not clear whether this receptor’
- Line 350: ‘whichlimits’ is now changed to ‘which limits’
Reviewer 2 Report
1. What is the main question addressed by the research? The aim of this study was to examine the contribution of microRNA (miRNA; miR) sequences, either delivered directly or modulated endogenously to these protective functions. The work is interesting and contribution to the field.
2. Do you consider the topic original or relevant in the field? Does it address a specific gap in the field? This work is review, summarizes the results of several studies and provides a clearer view of the issue.
3. What does it add to the subject area compared with other published material? CVD is still the leading cause of death worldwide and I find it very beneficial to look for any way to reduce it. Uncovering the possible mechanisms by which HDL can modulate the cellular miRNA landscape may provide potential new avenues for therapeutic intervention.
4. Are the conclusions consistent with the evidence and arguments presented and do they address the main question posed? The conclusions are adequate and correspond to the stated objectives of the work.
5. Are the references appropriate? Yes, the references are appropriate.
6. Please include any additional comments on the tables and figures. The presentation of some data in graphical or tabular form can provide better readability of the results.
Author Response
I thank the reviewer for their support, and the useful suggestion of presenting some of the data in a graphical or tabular form to aid readability of the manuscript. To address this point, we have now introduced two new schematic figures (Figure 1 and Figure 2) to support the text.
Reviewer 3 Report
Minor points:
Although the review is well documented, some sections are difficult to read, because the paragraphs are too long. For example: section: 2.3, line 126 to 131 or 137 to 142. This section needs to be redone
2.- I would also recommend including figures in the sections, because can help the reader
3.- Check the abbreviation in all the text
4.- Some punctuation marks are missing (line 89) or line 216, line 306. Please, check all the text.
5.- Finally, Some sentence must be redone, such as: line 320-321
Author Response
I thank the reviewer for the time and care taken to peer review this manuscript. My responses are highlighted within the manuscript for convenience of checking.
- The reviewer comments that, in places, the text is difficult to read due to paragraph length, and exemplifies Section 2.3, line 126-131 and line 137-142.
Sentence (126-131) is indeed far too long, and is now amended to three shorter sentences thus:
‘The ABCA1 and ABCG1 transporters also play important roles in the regulation of adaptive immune responses mediated by T-cells. T-cell receptor responses are strongly influenced by the composition and lipid raft structure of the plasma membrane (reviewed in [32]). T-cell activation decreases expression of ABCA1 and ABCG1, which is associated with the activation of protein tyrosine kinases Lck and Zap 70, and activation of MAP and ERK1/2 kinases [33].’
Sentence (137-142) also has the same problem, and is now amended to:
‘Cholesterol homeostasis via ABCG1 is also central to the role of regulatory T-cells (Tregs), which suppress innate and adaptive immune responses [36]. In the absence of the ABCG1 transporter, there is a selective expansion of Tregs [36], that reflects the role of elevated cholesterol. In turn, this inhibits mammalian Target of Rapamycin (mTOR) signalling, and favours STAT5 (signal transducer and activator of transcription) signalling.’
- The reviewer recommends using figures in the sections.
- The abbreviations throughout the text should be checked: this has now been done, with all added full names for abbreviations highlighted throughout the text.
- Some punctuation marks are missing: these have been amended (lines 89, 216, 306).
- Some sentences must be redone (e.g. lines 301-321).
With regard to points ‘1’, ‘4’ and ‘5’: the entire text (including lines 301-321 in the original text) has been tested using ‘Spelling and Grammar check’ (Word), to resolve any punctuation issues, and all long sentences identified have been eliminated and/or simplified (highlighted).
Reviewer 4 Report
The topic of the review is modulation of the cellular microRNA landscape by HDL and the potential contribution of this modulation to protection conferred by HDL. The review is well-written.
1. HDL are multimolecular platforms and native HDL carries several microRNAs. Besides discussing the lipidome and the proteome of HDL, this aspect of HDL biology should be mentioned.
2. In general, a lot is known on the biology of HDL after more 6 decades of research on these particles. In contrast, clinical translation of HDL-targeted therapies has not been successful. The results of the AEGIS-II study are being awaited. Several areas outside atherosclerotic cardiovascular disease have been considered e.g. diabetic wound healing and heart failure with preserved ejection fraction. A paragraph on clinical translation should be added.
Author Response
- The reviewer asks that the microRNA cargo of HDL be discussed.
We have now included some additional material (revised text lines 284-288) and used clearer signposting so that this content should be easily accessible to the reader (highlighted). In brief, this content is signposted at the end of Section 1.1 (revised text, lines 88-90, directing the reader to Section 2.2 and Section 3), and Section 2.2 has been re-titled ‘HDL microRNA cargo: cellular export and import of microRNA to HDL’. New material has been included (revised text, lines 284-288); we have also updated the text to include some new thoughts on the physio-chemical interactions of HDL with small RNA from Michell and coworkers (2022) (revised text lines 303-305).
- The reviewer suggests including a paragraph on clinical translation.
We have now added an additional paragraph, prior to the Conclusion, addressing this point (revised manuscript, lines 565-587).